# Flattening the Curve after the Initial Outbreak of Coronavirus Disease 2019: A Data-Driven Modeling Analysis for the Omicron Pandemic in China

**DOI:** 10.3390/vaccines11051009

**Published:** 2023-05-22

**Authors:** Jiaqi Sun, Yusi Li, Lin-Fan Xiao, Ning-Yi Shao, Miao Liu

**Affiliations:** 1Department of Mathematics, Faculty of Science and Technology, University of Macau, Taipa, Macau, China; 2Department of Biomedical Sciences, Faculty of Health Sciences, University of Macau, Taipa, Macau, China; 3MoE Frontiers Science Center for Precision Oncology, University of Macau, Taipa, Macau, China; 4Department of Pathology, Brigham and Women’s Hospital, Harvard Medical School, Boston, MA 02115, USA

**Keywords:** COVID-19, data-driven modeling analysis, flatten-the-curve policy

## Abstract

China is relaxing COVID-19 measures from the “dynamic zero tolerance” (DZT) level. The “flatten-the-curve” (FTC) strategy, which decreases and maintains the low rate of infection to avoid overwhelming the healthcare system by adopting relaxed nonpharmaceutical interventions (NPIs) after the outbreak, has been perceived as the most appropriate and effective method in preventing the spread of the Omicron variant. Hence, we established an improved data-driven model of Omicron transmission based on the age-structured stochastic compartmental susceptible-latent-infectious-removed-susceptible model constructed by Cai to deduce the overall prevention effect throughout China. At the current level of immunity without the application of any NPIs, more than 1.27 billion (including asymptomatic individuals) were infected within 90 days. Moreover, the Omicron outbreak would result in 1.49 million deaths within 180 days. The application of FTC could decrease the number of deaths by 36.91% within 360 days. The strict implementation of FTC policy combined with completed vaccination and drug use, which only resulted in 0.19 million deaths in an age-stratified model, will help end the pandemic within about 240 days. The pandemic would be successfully controlled within a shorter period of time without a high fatality rate; therefore, the FTC policy could be strictly implemented through enhancement of immunity and drug use.

## 1. Introduction

Since the COVID-19 outbreak occurred in late December 2019, the severe acute respiratory syndrome coronavirus 2 (SARS-CoV-2) has constantly mutated into multiple variants that have been identified to date [1]. The Omicron variant has been the dominant strain circulating worldwide [2]. Even in China, despite the strict implementation of massive anti-epidemic measures, Omicron flare-ups occurred occasionally; therefore, the nationwide surge of Omicron is still imminent [3]. Although current data suggests that Omicron has a surprisingly high rate of transmission, rapid replication in the human bronchus, and strong immune escape potential, it does not seem to cause more severe disease compared with other SARS-CoV-2 variants [4]. The high rate of asymptomatic infection, mild impairment of the lungs and other organs, and short recovery time put into doubt the necessity of epidemic prevention and control with strict implementation of nonpharmaceutical interventions (NPIs) [5,6,7].

The low infection rate in China was attributed to the continuous adoption of the dynamic zero-tolerance policy [8]. In the ongoing Omicron-dominant pandemic, the specific policy that should be applied remains controversial. In some countries, the negative impact of COVID-19 has somehow been reduced; however, although lower infection rates were achieved, the zero-COVID policy is still applied in China [9]. Completely eradicating the pandemic within the shortest possible time is the safest method to protect the older population, as the application of high-risk policies will further prolong the pandemic. According to previous mathematical model research, although >90% of the population in China had already been vaccinated, more than 1.5 million people will die of COVID-19 if all NPIs are discontinued while still confronting Omicron transmission; considering the Chinese population’s current immune level, the majority are expected to develop the infection within 6 months [10]. Totally no intervention (TNI) strategy would put enormous pressure on the medical system. Therefore, a less stringent, non-lockdown, but effective strategy is required to curb the transmission of Omicron; however, the stringency between DZT and TNI is considered to compromise the solution. Typically, the number of infected people in the COVID-19 outbreak suddenly increases during the initial period, which is illustrated as a steep peak-like curve in a statistical graph. The implementation of mild NPI measures can decrease the current infection rates; hence, the curve could be flattened for a longer time, and the number of cases will gradually decline, resulting in a smaller number of new infection cases. This degree of prevention was referred to as the “flatten-the-curve” response to the pandemic, which permits virus transmission at a low level without affecting the overall healthcare system [11,12]. Unduly harsh policies may cause inconveniences and significant financial burdens. China has the highest proportion of older adults among the countries worldwide, and a large part of this population with underlying health conditions are more vulnerable to COVID-19 [13,14,15,16]. Therefore, total liberalization of intervention programs to control the pandemic is suspected to increase the risk of severe disease or death from the Omicron virus. Compared with the “all-or-none” response strategy, the FTC policy aims to maintain social homeostasis, avoid the possible collapse of the healthcare system, and reduce overall mortality [11,12].

Whether the FTC policy is the most effective strategy to cope with the Omicron epidemic, the kinds of drugs, and level of population protective immunity necessary to stop the pandemic remains unclear. In this study, we established an improved data-driven model of Omicron transmission based on the age-structured stochastic compartmental susceptible-latent-infectious-removed-susceptible model constructed by Cai to evaluate the overall preventative effect of adopting the FTC policy in future pandemics in China.

## 2. Methods

### 2.1. SARS-CoV-2 Omicron Transmission, Vaccination Rate, and Disease Burden

The baseline model used in our study was improved based on the age-structured stochastic compartmental susceptible-latent-infectious-removed-susceptible model developed by the research team from Shanghai, China [10]. In the baseline scenario, five imported infections were used in the baseline simulations performed for 12 months. Then, 20 imported infections were included in the sensitivity analysis (Appendix A). The basic reproduction number (R) of the Omicron subtype was estimated to be more than 3-fold that of the Delta variant (from 3 to 8) [17,18]. The improvements in our model were as follows: (1) the effect of herd immunity was considered in our model, as the immunity was (*x* + 1)^2^, with *x* being the infected population; (2) the scenario regarding the reduction of effective contacts with different intensity levels among different age groups was analyzed.

Then, the number of infections, confirmed cases, hospitalizations, intensive care unit (ICU) admissions, and deaths were calculated to assess the SARS-CoV-2 Omicron burden. The parameters used in the mathematical model for inferring the above index on SARS-CoV-2 Omicron progression were the same as those used by the Shanghai research team [10]. With regard to healthcare resources, a total of 9.1 million hospital beds were utilized; of these, 3.14 million beds were allocated for patients with respiratory illnesses admitted in the internal medicine department, pediatric department, infectious disease department, and ICUs in China, of which 64,000 belonged to the ICU [19].

### 2.2. Mitigation with Vaccination, Antiviral Therapies, and NPIs

Inactivated vaccines were administered, and the proportion of vaccination in the different age groups was the same as that reported by the Shanghai research team [10]. The vaccine efficacy on infection, onward transmission rate, symptoms, hospitalization rate, vaccination transition rate, and recovery rate were the same as those reported in a previous study [10]. We adjusted the vaccine effectiveness against COVID-19-related deaths, as shown in Appendix A.

In the baseline scenario, no antiviral therapies were provided to the patients with confirmed cases. To quantify the mitigation effect of antiviral therapies, we simulated another scenario: approximately 75% of patients with confirmed cases were treated with Paxlovid in accordance with the Diagnosis and Treatment Protocol for Novel Coronavirus Pneumonia (Trial Version 7) issued by the Chinese Center for Disease Control and Prevention [20]. The effectiveness rate of the antiviral therapies was assumed to be 75%.

The different NPI intensity levels (NPIIL) at specific timepoints to avoid a shortage of ICU beds were measured. The NPIIL was rated from 0 to 1. An NPIIL level of 0.01 indicated that the strictest NPI was implemented, while level 1 indicated that no NPI was implemented. In the baseline scenario, no NPI was carried. To quantify the mitigation effect of NPIs, we simulated two alternative scenarios: (1) reducing the effective contacts with the same intensity level among different age groups; and (2) reducing the effective contacts with different intensity levels among different age groups.

To evaluate the effect of the positive scenario, we used the Singapore scenario as a reference; the completed mRNA vaccine efficacy is shown in Appendix A. The definition of completed vaccine was adopted from the standards of the Ministry of Health in Singapore [21]. In the positive scenario, the proportion of older adults aged ≥60 years who could not receive vaccination was only 0.1. Hence, we assumed that 75% of patients with confirmed cases received Paxlovid. The SARS-CoV-2 Omicron burden was predicted at specific times among different age groups using different NPIIL and NPI mitigation strategies based on the positive scenario (Appendix A).

## 3. Statistical Analysis

A total of 200 stochastic simulations were performed for each scenario. The results of these simulations determined the distribution of infections, confirmed cases, hospitalizations, ICU admissions, and deaths by age. Then the median, the 2.5% quantile, and 97.5% quantile of 200 simulations were calculated.

## 4. Data Availability and Code Availability

All data used in this modeling study were collected from publicly available databases (https://www.moh.gov.sg/covid-19/statistics (accessed on 1 December 2022), https://www.dsec.gov.mo/zh-MO/Statistic?id=1 (accessed on 10 October 2022), https://covid-19.ersinfotech.com/stats (accessed on 10 October 2022), https://www.nature.com/articles/s41591-022-01855-7 (accessed on 1 August 2022)). The codes are available on GitHub (https://github.com/shaolab-UM/Flatten_the_curve_Omicron_transmission_Model (accessed on 10 December 2022)).

## 5. Results

### 5.1. Baseline Scenario

In the baseline scenario, NPI protocols for the prevention of SARS-CoV-2 Omicron transmission were not applied, and antiviral therapies for controlling the progression of the disease were not administered. The simulated conditions in our model were based on the age-structured stochastic compartmental susceptible-latent-infectious-removed-susceptible model of SARS-CoV-2 transmission. The different values presented between our model and the previous model indicate the transmission rate; moreover, herd immunity was introduced in our model: (1) the transmission rate in the absence of NPIs, which was inferred from the reproduction number of the SARS-CoV-2 Omicron variant, was set at 7.2; and (2) herd immunity was determined based on the proportion of daily infectious cases.

The predicted daily SARS-CoV-2 Omicron burden in China according to the baseline scenario is shown in Figure 1. The total number of infectious cases and accumulated number of deaths in the total population were 1273.27 million (close to 1.27 billion at 90 days) and 1.49 million, respectively. The total number of infectious cases and accumulated number of deaths among the older group were 244.60 million and 1.40 million, respectively, accounting for 19.21% of the total number of infectious cases and 93.97% of the total accumulated deaths. On day 52, the total daily infectious cases reached 92.29 million, among which 17.56 million (19.03%) were older adults. The total daily deaths peaked on day 59 (72,093), 68,122 (94.45%) of whom were older adults. On day 58, the numbers of non-ICU patients and ICU patients surged to 1.67 million and 1.07 million, respectively. Moreover, 1.05 million and 0.98 million older adults required hospital and ICU admission, respectively, accounting for 62.64% and 91.57% of the total number of non-ICU and ICU patients, respectively. From day 44 to day 88, the required number of ICU beds exceeded the number of ICU beds available in China.

### 5.2. Classification of the Stringency of NPIs

The NPI intensity was rated on a scale of 0–1. An NPIIL of 0.01 indicated that the strictest NPI was carried out, while an NPIIL of 1 indicated that no NPI was implemented. Our baseline scenario model is used to predict the daily number of infectious cases caused by the SARS-CoV-2 Omicron variant in Macau. Here, we presumed that five initial infections were introduced. The daily predictive and actual SARS-CoV-2 Omicron burdens in Macau are shown in Appendix A. After comparing the real-world data with the predicted data, we determined the specific NPIs of the NPIIL in the model that correspond to the real-world situation. The application of FTC could decrease deaths by 36.91% within 360 days. (Table 1 and Appendix A).

### 5.3. Impact of NPI Mitigation Strategies

We carried out different NPIIL and NPI mitigation strategies at specific timepoints to avoid the shortage of ICU beds. The following two subset scenarios were separately analyzed: (1) all patients with confirmed cases did not receive drug treatment; and (2) 75% of patients with confirmed cases received drug treatment. The efficacy of the drug was assumed to be 75%.

Results of the prediction of daily ICU beds required using different NPIIL and NPI mitigation strategies at specific timepoints are shown in Figure 2. In all patients with confirmed cases who did not receive Paxlovid, the NPIs were carried out on days 1, 40, 170, and 270 with NPIILs of 0.8, 0.2, 0.3, and 0.5, respectively (Figure 2A). The number of daily ICU beds required peaked on day 127 (65), and a total of 65,197 ICU beds were needed (Figure 2A). For older patients, the daily ICU beds needed on day 127 were 60,923 (93.46%) (Figure 2A). The total number of infections in 360 days was 680.40 million, of whom 134.45 million (19.76%) were older patients (Figure 2B). The total number of deaths in 360 days was 0.94 million, of whom 0.89 million (95.25%) were older patients (Figure 2C).

In 75% of the patients with confirmed cases who received Paxlovid, the NPIs were carried out on days 1, 45, 100, and 160 with NPIILs of 0.8, 0.2, 0.3, and 0.7, respectively (Figure 2E). The daily ICU bed need peaked on day 192 (71,037 ICU beds) (Figure 2E). For older patients, the daily ICU beds needed on day 192 were 64,577 (90.91%) (Figure 2E). The total number of infections in 360 days was 1053.20 million, of whom 204.25 million were older patients (19.39%) (Figure 2F). The total number of deaths in 360 days was 0.58 million, of whom 0.55 million (94.44%) were older patients (Figure 2G).

### 5.4. Impact of NPI Mitigation Strategies by Age

We investigated the predicted daily SARS-CoV-2 Omicron burden with different NPIILs of NPI mitigation strategies at specific timepoints among different age groups under the following two conditions: (1) all the confirmed cases did not receive Paxlovid, and (2) 75% of the patients with confirmed cases received Paxlovid.

Different NPIILs of NPI mitigation strategies at specific timepoints among different age groups were set to avoid the shortage of ICU beds. The NPIIL was separately defined in three contact groups: “inside <60 group (contact between the population aged <60 years)”, “between group (contact between the population aged <60 years and the older adults aged ≥60 years)”, and “inside ≥60 group (contact between older adults aged ≥60)”.

In all patients with confirmed cases who did not receive Paxlovid, the NPIs were conducted on days 1, 14, 45, 150, 220, and 270 with different NPIILs in the “inside < 60 group”, “between group”, and “inside ≥ 60 group” (Figure 3A). The number of daily ICU beds needed peaked on day 200 (66,020 ICU beds) (Figure 3A). For older patients, the daily ICU beds needed on day 200 were 54,728 (82.90%) (Figure 3A). The total number of infections in 360 days was 990.67 million, of whom 103.59 million (10.46%) were older adults (Figure 3B). The total number of deaths in 360 days was 0.81 million, of whom 0.74 million (90.88%) were older adults (Figure 3C).

In 75% of the patients with confirmed cases who received Paxlovid, the NPIs were conducted on days 1, 14, 45, and 120 with different NPIILs in the “inside < 60 group”, “between group”, and “inside ≥ 60 group” (Figure 3E). The daily ICU beds peaked on day 94 (62,888 ICU beds) (Figure 3E). In older patients, the daily ICU beds needed on day 94 were 54,798 (87.14%) (Figure 3E). The total number of infections in 360 days was 1019.30 million, of whom 119.39 million (11.71%) were older patients (Figure 3F). The total number of deaths in 360 days was 0.40 million, of whom 0.37 million (91.47%) were older patients (Figure 3G).

### 5.5. Comparison of the Outcomes of Different Strategies

We also assumed an ideal scenario in which different age-level groupings of NPI were implemented, 75% of the patients took Paxlovid, and the mortality rate decreased to 0.025% after vaccination (based on data obtained from Singapore). We explored the accumulated number of infections and deaths in the total population and older groups following the use of different strategies.

The accumulated infections and accumulated deaths of the total population and older adults in 360 days in different models are shown in Figure 4 and Table 2. In Table 2, the prevalence and fatality of SARS-CoV-2 Omicron in the total population and older adults were also calculated. In Model 5, in which 75% of patients with confirmed cases received Paxlovid and different NPIILs of NPI mitigation strategies were applied among different age groups at specific timepoints to avoid a shortage of ICU beds, the total number of infectious cases was 1.41 billion, accounting for 72.19% of the total number of people in the country; meanwhile, the total number of infectious older adults was 267.36 million, accounting for 44.66% of the total number of older people in the country. The fatality rates were 0.04% in the total population and 0.31% in the older group. When the restrictions were fully lifted after 180 days, 170,966 people died, of whom 159,268 were older adults. In the ideal model, the prevalence rates of Omicron infection were 71.66% in the total population and 59.67% in older adults. The fatality rates in the total population and older adults were 0.02% and 0.11%, respectively. When the restrictions were fully lifted 180 days later, 61,957 people died after 180 days, of whom 51,909 were older adults.

The greatest number of fatal falls occur if the baseline status peaks within a short time. The least number of deaths was reported in the ideal scenario; hence, we recommend the following model: “Paxlovid treatment in 75% of the population + application of age-stratified FTC + receipt of complete vaccine (3 doses of mRNA) in 90% of the population”. The number of deaths in this model was less than 200,000 when the peak value was reached (within 240 days); hence, these findings show that the pandemic has a tendency to end within a shorter period of time. In other models, the number of deaths reported was several times greater.

## 6. Discussion

Although the transmission model based on Shanghai’s data made some predictions about the Omicron transmission in China, some of the conclusions in this study were not applicable to the real-world situation. According to their inferences, the total number of infected people was significantly higher compared with that of the symptomatic people (more than 2 billion), which is beyond the actual population size [10]. In the current study, we corrected and optimized the modelling system and followed the epidemiological data reports in Macau, Hong Kong, and Singapore in 2022. We also determined whether the FTC policy was an effective method for the prevention and control of the Omicron outbreak. Here, we modified the model to revise the effect of the herd immunity (see Section 2), as the revised model indicated that the basal scenario will reach 1.27 billion accumulated patients at day 180. Our results also proved that, depending on the kind of FTC prevention applied, several hundred thousand deaths may occur. According to the conclusions of other studies, the insufficient immunity level of the Chinese population was one of the reasons for the continued implementation of the DZT policy [10]. COVID-19 mixed vaccines or bivalent COVID-19 vaccines contribute to the significant increase in antibody levels and provide broad protection for most of the populations; therefore, multiple sources of vaccines are an important prerequisite to ending the pandemic [22,23]. The promotion of immunity through vaccination along with the implementation of FTC policy would effectively overcome the COVID-19 outbreak within a shorter period of time.

The implementation of FTC policy permits virus transmission at a relatively low level; however, the Omicron variant is one of the most highly infectious viruses; although the case fatality rate was close to or lower than that of influenza, the high number of infected individuals would result in an increased number of deaths [10]. Promisingly, the Omicron variant has low pathogenicity; therefore, patients with asymptomatic cases and mild symptoms do not require hospital admission to avoid exhausting the healthcare system. In outbreaks caused by a virus with extremely high transmissibility, such as Omicron, the implementation of non-DZT policies is not effective in preventing and controlling its spread among individuals of all ages. The national health code system for every citizen had been established in China, which provided data on age, sex, and movement of people in real time [24]. This system contributes to the implementation of an age-stratified FTC. However, this study has some limitations. China had increased the ICU bed capacity, but we only used 64,000 beds in this study. Owing to the difference in vaccination types, the actual number of deaths may have been underestimated. In our research, FTC should be combined with Paxlovid use and mRNA vaccination in order to reduce the number of deaths. Paxlovid is an oral drug for patients with mild COVID-19 to avoid progression to severe COVID-19, which requires hospitalization [25]. The scope of application (75%) in our model is reasonable. Although 90% of the population received inactivated vaccines, with the development of SARS-CoV-2 mutants, mRNA, especially bivalent mRNA vaccines, showed a more broad and strong protective function against Omicron [26]. Omicron outbreak would result in 1.49 million deaths within 180 days if NPI protocols for the prevention of SARS-CoV-2 Omicron transmission were not applied and antiviral therapies for controlling the progress of the disease were not administered (Model 1). The application of FTC could decrease deaths by 36.91% within 360 days. The strict implementation of FTC policy combined with completed vaccination and drug use (Model 7), which only resulted in 0.19 million deaths in an age-stratified model. Therefore, the combination of FTC, drug use, and enhanced vaccination would play an important role, especially in future potential outbreaks.

## 7. Conclusions

FTC can decrease deaths as an anti-pandemic, precise prevention method targeting the older population during the initial outbreak of Omicron and could be a cost-effective strategy, along with enhanced immunity levels and drug use. Therefore, more ICU beds, highly effective vaccines, and Paxlovid drugs should be prepared as part of the implementation of the FTC strategy.

## Figures and Tables

**Figure 1 vaccines-11-01009-f001:**
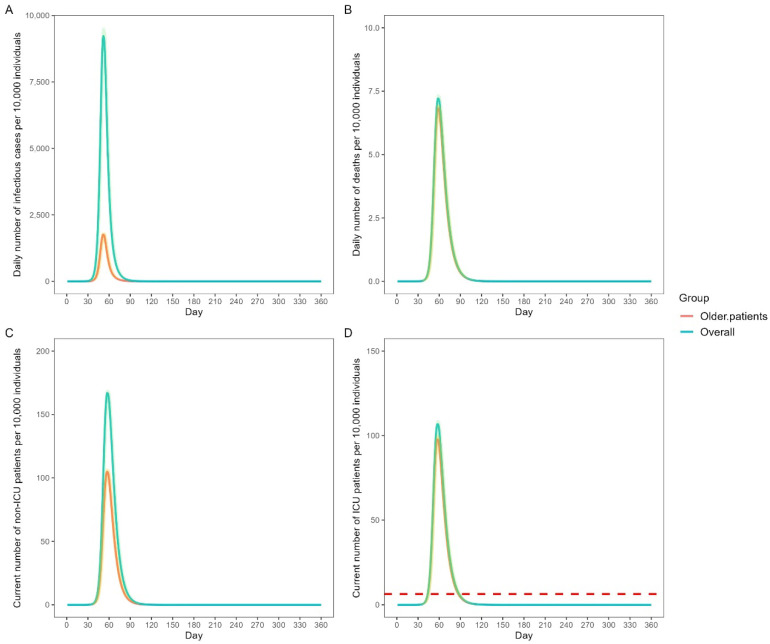
Predicted SARS-CoV-2 Omicron burden in China with the baseline scenario. (**A**) Daily number of infectious cases per 10,000 individuals; (**B**) Daily number of deaths per 10,000 individuals; (**C**) Current number of non-ICU patients per 10,000 individuals; (**D**) Current number of ICU patients per 10,000 individuals with the red dashed line indicating the number of ICU beds available in China. All data are presented as medians with 2.5% and 97.5% quantiles for n = 200 simulations. ICU: intensive care unit.

**Figure 2 vaccines-11-01009-f002:**
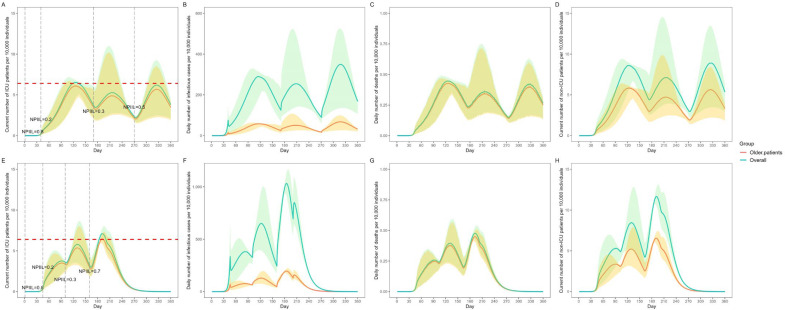
Prediction of SARS-CoV-2 Omicron burden using different NPIILs of NPI mitigation strategies at specific timepoints. (**A**) Current number of ICU patients per 10,000 individuals who did not receive Paxlovid; (**B**) Daily number of infectious cases per 10,000 individuals who did not receive Paxlovid; (**C**) Daily number of deaths per 10,000 individuals who did not receive Paxlovid; (**D**) Current number of hospitalized (non-ICU) patients per 10,000 individuals who did not receive Paxlovid; (**E**) Current number of ICU patients per 10,000 individuals with 75% of patients with confirmed cases receiving Paxlovid; (**F**) Daily number of infectious cases per 10,000 individuals with 75% of patients with confirmed cases receiving Paxlovid.; (**G**) Daily number of deaths per 10,000 individuals with 75% of patients with confirmed receiving Paxlovid.; (**H**) Current number of hospitalized (non-ICU) patients per 10,000 individuals with 75% of patients with confirmed cases receiving Paxlovid. In (**A**,**E**), the red dashed line indicates the number of ICU beds available in China. NPIIL: nonpharmaceutical intervention intensity level; NPIILs ranged between 0 and 1. AN NPIIL level of 0.01 indicates that the strictest NPI was carried out, while an NPIIL level of 1 indicated that no NPI was implemented. All data are presented as medians with 2.5% and 97.5% quantiles for n = 200 simulations.

**Figure 3 vaccines-11-01009-f003:**
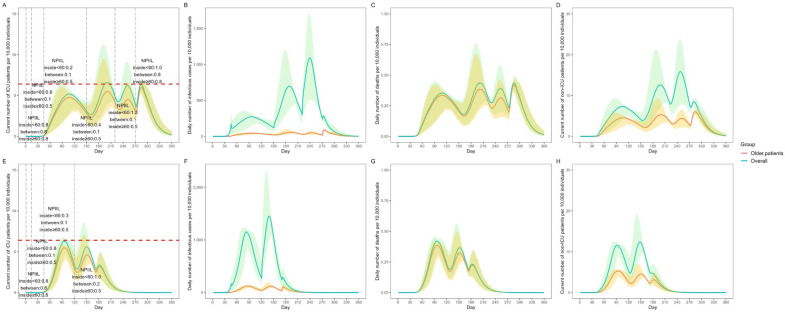
Prediction of SARS-CoV-2 Omicron burden using different NPIILs of NPI mitigation strategies at specific timepoints among different age groups. (**A**) Current number of ICU patients per 10,000 individuals who did not receive Paxlovid; (**B**) Daily number of infectious cases per 10,000 individuals who did not receive Paxlovid. (**C**) Daily number of deaths per 10,000 individuals who did not receive Paxlovid; (**D**) Current number of hospitalized (non-ICU) patients per 10,000 individuals who did not receive Paxlovid. (**E**) Current number of ICU patients per 10,000 individuals with 75% of patients with confirmed cases receiving Paxlovid; (**F**) Daily number of infectious cases per 10,000 individuals with 75% of patients with confirmed cases receiving Paxlovid; (**G**) Daily number of deaths per 10,000 individuals with 75% of patients with confirmed cases receiving Paxlovid; (**H**) Current number of hospitalized (non-ICU) patients per 10,000 individuals with 75% of patients with confirmed cases receiving Paxlovid; In (**A**,**E**), the red dashed line indicates the number of ICU beds available in China. NPIIL: nonpharmaceutical intervention intensity level; NPIILs ranged between 0 and 1. An NPIIL level of 0.01 indicates that the strictest NPI was carried out, while an NPIII level of 1 indicated that no NPI was implemented. Inside <60 group: contact between the population aged < 60 years, between: contact between the population aged <60 years and older adults aged ≥60 years; inside ≥60: contact between older patients aged ≥60 years; All data are presented as medians with 2.5% and 97.5% quantiles for n = 200 simulations.

**Figure 4 vaccines-11-01009-f004:**
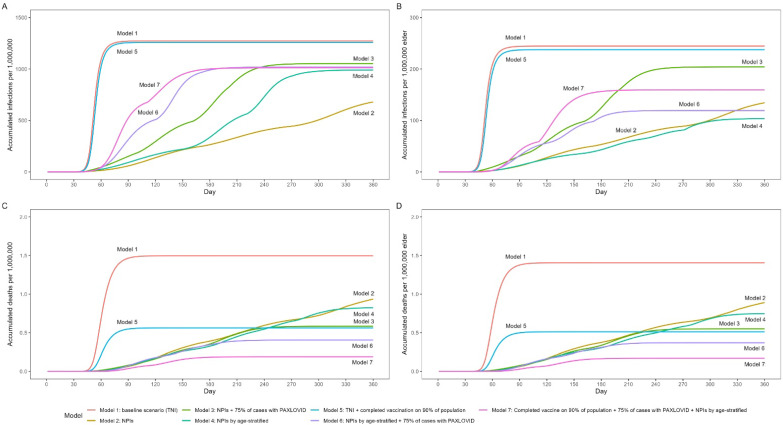
Comparison of the accumulated number of infectious cases and the accumulated number of deaths caused by SARS-CoV-2 Omicron in different models. (**A**) The overall accumulated number of infectious cases of SARS-CoV-2 Omicron in different models; (**B**) The accumulated number of infectious cases caused by SARS-CoV-2 Omicron among older adults in different models; (**C**) The overall accumulated number of deaths caused by SARS-CoV-2 Omicron in different models; (**D**) The accumulated number of deaths caused by SARS-CoV-2 Omicron among older adults in different models.

**Table 1 vaccines-11-01009-t001:** Classification of the NPIIL.

NPIIL	NPIs
0.80	Wearing a facemask
0.30	Banning dining in a restaurant and closing entertainment venues and other public places
0.05	Restrictions on going outdoors
0.02	Requirement for rapid antigen testing on a daily basis and quarantining of patients with confirmed cases

NPIs: nonpharmaceutical interventions. NPIIL: nonpharmaceutical intervention intensity level. NPIILs between 0 and 1. NPIII level 0.01 indicates that the strictest NPI was carried out, while NPIII level 1 indicates that no NPI was implemented.

**Table 2 vaccines-11-01009-t002:** Accumulated number of infectious cases and accumulated number of deaths caused by SARS-CoV-2 Omicron in different models.

Model	Overall		Older Patients	
	Accumulated Number of Infectious Cases	Accumulated Number of Deaths	Accumulated Number of Infectious Cases	Accumulated Number of Deaths
Model 1: baseline scenario (TNI)	1273.27 (90.16%)	1.49 (0.12%)	244.60 (91.49%)	1.40 (0.57%)
Model 2: NPIs	680.40 (48.19%)	0.94 (0.14%)	134.45 (50.29%)	0.89 (0.66%)
Model 3: NPIs + 75% of cases with Paxlovid	1053.20 (74.59%)	0.58 (0.06%)	204.25 (76.40%)	0.55 (0.27%)
Model 4: NPIs stratified by age	990.67 (70.16%)	0.81 (0.08%)	103.59 (38.75%)	0.73 (0.70%)
Model 5: TNI + completed vaccination on 90% of the population	1259.43 (89.19%)	0.56 (0.04%)	237.69 (88.90%)	0.51 (0.21%)
Model 6: NPIs stratified by age + 75% of cases with Paxlovid	1019.30 (72.19%)	0.40 (0.04%)	119.39 (44.66%)	0.37 (0.31%)
Model 7: completed vaccination on 90% of the population + 75% of cases with Paxlovid + NPIs stratified by age	1011.87 (71.66%)	0.19 (0.02%)	159.53 (59.67%)	0.17 (0.11%)

The population of China is 1.41 billion, and the total number of elders is 267.36 million.

## Data Availability

Data derived from public domain resources.

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
