# Peer review of "Flattening the Curve after the Initial Outbreak of Coronavirus Disease 2019: A Data-Driven Modeling Analysis for the Omicron Pandemic in China"

_vaccines, 2023, doi:10.3390/vaccines11051009_

Round 1
Reviewer 1 Report
Overall, the study presents a study that aims to evaluate the effectiveness of the "flatten-the-curve" (FTC) strategy in preventing the spread of the Omicron variant in China. The study uses an improved data-driven model to simulate the transmission of the virus and assesses the impact of various interventions, including nonpharmaceutical interventions (NPIs) and vaccination.
The study provides a clear and concise summary of the study. However, there are a few areas where additional information would be helpful. For example, it would be useful to know more about the specific NPIs that were considered and the assumptions underlying the model. Additionally, some context on the current state of the pandemic in China and the rationale for relaxing the COVID-19 measures would provide useful background information for readers.
1. The main question addressed by this research: Although the authors state to “establish a data-driven model, based on the progress of Omicron outbreak in Macau, Hong Kong, and Singapore in 2022, to evaluate the possibility of adopting the FTC policy in future pandemics in China,” there is no specific, clear research question or objective.
2. The topic is very important and relevant. The study addresses an important gap in the literature, considering its dire social and economic consequences.
3. This study uses a data-driven model, and this is a unique contribution – many studies rely on regression analysis that is largely driven by theory.
4. Data description is very limited, perhaps this section can be expanded to include a more thorough description.
5. Results are well presented; however, introduction may be better developed so that these results can be situated in an appropriate manner.
6. References are appropriate.
7. No other comments.
Reviewer 2 Report
Review of the Article:
Flattening the curve after the initial outbreak of coronavirus disease 2019: A data-driven modeling analysis for Omicron 3 pandemic in China.
The authors mentioned that China is moving away from its strict COVID-19 measures towards the "flatten-the-curve" (FTC) strategy, which aims to maintain low infection rates to avoid overwhelming the healthcare system. A new data-driven model has been established to study the prevention effect of the Omicron variant. They mention that without any nonpharmaceutical interventions (NPIs), over 1.27 billion individuals (including asymptomatic cases) would be infected within 90 days, and 1.49 million deaths would occur within 180 days. However, implementing FTC policies could reduce deaths by 36.91% within 360 days. Strictly implementing FTC policies, along with completed vaccination and drug use, would result in only 0.19 million deaths in an age-stratified model and could help end the pandemic within about 240 days, without resulting in a high fatality rate. Therefore, they support the idea that the FTC policy could be strictly implemented through immunity enhancement and drug use.
Methods:
The authors mention Figure S1 but I couldn’t find any supplementary figure. I’ve just had acces to Table S1.
Data and code availability:
In this section, the authors mention that “The data used in the study are provided in the supplementary materials”. But as I mentioned before I’ve just had acces to Table S1.
Classification of the stringency of NPIs
Based on table 1. The following statement could be wrong,
“The NPI intensity was rated on a scale of 0–1. A NPIIL 0.1 indicated that the strictest
NPI was carried out, while NPIIL 1 indicated that no NPI was implemented.”
Then in the table 1, there are values smaller than 0.1
Please explain better or rephrase.
Figure 4: I suggest changing the colors of the lines or add a legend on each since it is very hard to understand which model is which. For instance model 1 and 7 have almost the same color. Also, models 4 and 5.
Besides, the correct word is model, not modle.
These figures need more explanation. Specifically, How these models reach to 0 new cases after certain number of days. It is very unlikely from a Biological standpoint that a disease could reach complete eradication, to the point that no new death occur.
Regarding table 2. Accumulated number of infectious cases and accumulated number of deaths caused by SARS-CoV-2 Omicron in different models.
Since you are mentioning accumulated values, it is important to mention what would be the period of time. This table contradicts figure 4 since you arrive to 0 new cases after certain number of days but still you reported in table 2 a constant “accumulated number of deaths” and “accumulated number of infectious cases”.
Discussion:
I think this statement should perhaps coincide with any of your models, or at least specify the model that you are referring to.
“Therefore, the combination of FTC, drug use, and enhanced vaccination would play an important role especially in future potential outbreaks.”
The article is very interesting but some aspects need a better explanation.
1) Raw data used was not available to see and understand its structure.
2) The authors seemed to contradict themselves when they report certain accumulated values that then are not present in some figures.
I strongly recommend re-write certain areas of the article so that the models make more sense from a biological standpoint.
